# Modulation of SUR1 K_ATP_ Channel Subunit Activity in the Peripheral Nervous System Reduces Mechanical Hyperalgesia after Nerve Injury in Mice

**DOI:** 10.3390/ijms20092251

**Published:** 2019-05-07

**Authors:** Wing Luu, James Bjork, Erin Salo, Nicole Entenmann, Taylor Jurgenson, Cole Fisher, Amanda H. Klein

**Affiliations:** 1Department of Pharmacy Practice and Pharmaceutical Sciences, University of Minnesota, Duluth, MN 55812, USA; luuxx075@d.umn.edu (W.L.); salox099@d.umn.edu (E.S.); enten002@d.umn.edu (N.E.); tjjurgenson@gmail.com (T.J.); nfisher@d.umn.edu (C.F.); 2Department of Biomedical Sciences, Medical School Duluth, Duluth, MN 55812, USA; jbjork1@d.umn.edu

**Keywords:** neuropathy, K_ATP_ channels, SUR1, Kir6.2, analgesia, spinal nerve ligation

## Abstract

The ATP-sensitive K^+^ channel (K_ATP_) is involved in hypersensitivity during chronic pain and is presumed to be a downstream target of mu opioid receptors. Multiple subtypes of K_ATP_ channels exist in the peripheral and central nervous system and their activity may be inversely correlated to chronic pain phenotypes in rodents. In this study, we investigated the different K_ATP_ channel subunits that could be involved in neuropathic pain in mice. In chronic pain models utilizing spinal nerve ligation, SUR1 and Kir6.2 subunits were found to be significantly downregulated in dorsal root ganglia and the spinal cord. Local or intrathecal administration of SUR1-K_ATP_ channel subtype agonists resulted in analgesia after spinal nerve ligation but not SUR2 agonists. In ex-vivo nerve recordings, administration of the SUR1 agonist diazoxide to peripheral nerve terminals decreased mechanically evoked potentials. Genetic knockdown of SUR1 through an associated adenoviral strategy resulted in mechanical hyperalgesia but not thermal hyperalgesia compared to control mice. Behavioral data from neuropathic mice indicate that local reductions in SUR1-subtype K_ATP_ channel activity can exacerbate neuropathic pain symptoms. Since neuropathic pain is of major clinical relevance, potassium channels present a target for analgesic therapies, especially since they are expressed in nociceptors and could play an essential role in regulating the excitability of neurons involved in pain-transmission.

## 1. Introduction

Neuropathic pain is a common clinical problem encountered by many patients, especially those with diabetes, cancer, herpes zoster, nerve compression and various other conditions. It is estimated that neuropathic pain affects 6–8% of the general population, although this number has been suggested to be as high as 17% [1,2]. Neuropathic pain greatly decreases quality of life and the direct medical cost to patients can cost thousands of dollars [3]. Unfortunately, many currently available pharmaceuticals do not treat peripheral neuropathic pain effectively; therefore, the development of better treatment strategies is urgently required. One important group of ion channels needed to control and maintain neuronal excitability are potassium channels. The functional role of only a few potassium channels are known during chronic pain and increased activity of potassium channels within peripheral nociceptors or spinal cord pathways could provide pharmacological targets for new analgesics [4].

ATP-sensitive potassium (K_ATP_) channels are inwardly rectifying potassium channels that couple metabolism to electrical activity of cells. K_ATP_ channels are hetero-octamers composed of pore-forming Kir6.1 or Kir6.2 subunits and sulfonylurea receptor (SUR1 or SUR2) regulatory subunits (4:4). These channels are widely expressed in most metabolically active tissues, including the nervous system [5]. Previous studies have found that agonists or downstream activation of K_ATP_ channels have been implicated in the reduction of acute and chronic pain in rodent models of skin incision [6], formalin injection [7] and spinal nerve ligation [8]. Unfortunately, most studies have not closely investigated which K_ATP_ channel subunits are contributing to the analgesic effects of these treatments or where these K_ATP_ channels are located (e.g., spinal cord versus peripheral nervous system). Previous data suggest that Kir6.2/SUR1 is the predominant functional subtype within the peripheral nervous system but pain sensitization appears to be mediated mostly by SUR2 after cutaneous injury [9,10,11,12,13,14]. Current data are also lacking on K_ATP_ channel expression changes occurring after initiation of chronic pain in the peripheral versus central nervous system over an extended (>30 days) period of time. This is important, as many animal models of chronic pain do not accurately reflect clinical pain syndromes due to the lack of time progression between injury and phenotypic assessment [15].

In this study, we show that *Abcc8* (SUR1) and *Kcnj11* (Kir6.2) gene expression is decreased in the dorsal root ganglia and sciatic nerves two months post spinal nerve ligation (SNL) in mice. Administration of SUR1-subtype K_ATP_ channel agonists, by intraplantar or intrathecal injection, were able to alleviate mechanical hypersensitivity after SNL. Mice lacking the SUR1 K_ATP_ channel subunits due to genetic modification, either by global knockout (SUR1 KO) or intrathecal injection of short hairpin RNA (*Abcc8* shRNA), displayed mechanical hypersensitivity when compared to wild type and control mice. Taken together, these data suggest that K_ATP_ channel expression and function make an important contribution to the development of neuropathic pain.

## 2. Results

### 2.1. Gene Expression of SUR1 and Kir6.2 Decreases in the Peripheral Nervous System after Spinal Nerve Ligation

It has been previously shown that different K_ATP_ channel subunits are downregulated during chronic pain [9,12]. We attempted to confirm this by comparing the ipsilateral and contralateral dorsal root ganglia (DRG), sciatic nerves (SN) and spinal cords of mice after SNL. Expression of *Abcc8* and *Kcnj11* iso2 mRNA was significantly decreased in ipsilateral (i.e., injured) dorsal root ganglia versus contralateral (i.e., uninjured; Figure 1A,D). Abcc9 and *Kcnj11* iso1 were also largely downregulated in ipsilateral DRG (Figure 1B,C), but were not statistically significant- compared to contralateral DRG. For samples where matching ipsilateral and contralateral data were available, 70% of *Abcc8* and nearly all *Kcnj11* (100% for *Kcnj11* iso1 and 88.9% for *Kcnj11* iso2), the ipsilateral samples were lower than the contralateral DRG samples (Figure 1E). Similar data for *Abcc* and *Kcnj* subunits were also found in sciatic nerves (Figure 2A–D). In sciatic nerves, the number of ipsilateral samples for *Abcc8* and *Kcnj11* iso2 lower than the contralateral samples were 63.6% and 72.7%, respectively (Figure 2E). There were no significant expression differences across K_ATP_ channel subunits in the spinal cord (Figure 3A–E). However, the number of spinal cord samples where *Abcc8* and *Kcnj11* iso2 were decreased in ipsilateral versus contralateral spinal cord were 81.8% and 100%, respectively (Figure 3F). *Kcnj8* is not reported to be in the peripheral nervous system (PNS) [12] but the expression in the spinal cord was not altered after SNL (Figure 3E,F). Our own mRNA expression data also suggest that *Kcnj8* is mostly expressed in the spinal cord and brainstem and is not expressed in cell bodies of peripheral nerves, including the dorsal root ganglia and trigeminal ganglia (Appendix A).

### 2.2. Local Delivery of SUR1 Agonists Alleviate Mechanical Hypersensitivity after Spinal Nerve Ligation

After spinal nerve ligation, various K_ATP_ channel subtypes were pharmacologically stimulated or inhibited either through injection into the nerve lesioned hindpaw (i.e., intraplantar injection) or directly into the spinal subarachnoid space (i.e., intrathecal injection). Mechanical and thermal paw withdrawal testing were performed at several time points post injection to determine if reported analgesia by opening K_ATP_ channels could be modality-specific. Intraplantar injection of SUR1 agonists, diazoxide and NN414, increased mechanical paw withdrawal thresholds in spinal nerve ligated mice (repeated measures ANOVA, F (5, 44) = 9.719, * *p* < 0.05, CI diazoxide = −2.999 to −0.9227, CI NN414 = −1.167 to −0.6546; Figure 4A,E) compared to vehicle treated mice (Figure 4L). Interestingly, thermal thresholds were not significantly increased after intraplantar administration of SUR1 agonists (Figure 4B,F). Intraplantar injection of SUR2 targeting K_ATP_ channel agonists, including pinacidil, levcromakalim and nicorandil, did not significantly alter either thermal or mechanical withdrawal latencies/thresholds compared to vehicle (Figure 4C,D,G–J,M). Antagonists of SUR1/SUR2 subtype K_ATP_ channels were also tested to see if mechanical and thermal sensitivity would be altered (Figure 5A–F). Although there were numerical decreases in mechanical thresholds after administration of tolbutamide and glyburide (Figure 5A,C), these were not significantly different compared to vehicle treatment (Figure 4L). Mechanical and thermal paw withdrawal thresholds collected from the un-injected paw (i.e., contralateral paw) did not show significant changes in force thresholds or latencies [16].

Since intraplantar administration of K_ATP_ channel agonists appeared to alter mechanical sensitivity but do not appear to affect thermal sensitivity after SNL, only mechanical paw withdrawal testing was performed in behavioral experiments after intrathecal injection (Figure 6A–G). Intrathecal injection of SUR1 agonists, diazoxide and NN414, significantly increased mechanical paw withdrawal thresholds (repeated measures ANOVA, F (6, 56) = 32.94, * *p* < 0.05, CI diazoxide = −0.7470 to −0.4126, CI NN414 = −0.999 to −0.1607; Figure 6A,D) compared to vehicle (Figure 6G). Interestingly, the Kir6.2/SUR2 agonist levcromakalim also increased mechanical paw withdrawal thresholds after intrathecal injection (repeated measures ANOVA, CI levcromakalim = −3.363 to −1.969; Figure 6E). The analgesic effects of SUR1 agonists, particularly diazoxide, were apparent under both local and spinal administration. We suspect that the analgesia produced through administration of SUR1 agonists is most likely mediated through K_ATP_ channels, as mice globally lacking the SUR1 subunit have minimal analgesia after intraplantar injection (Appendix A).

### 2.3. Local Diazoxide Administration Alters General Movement in Open Field Testing

In order to determine if SUR1 agonists can improve animal motility after SNL, we measured the mobility and activity of mice using an open field test. Since diazoxide had a robust effect on mechanical hypersensitivity within 30 min post intraplantar injection (Figure 4), we decided to measure any changes in overall activity compared to vehicle injection in SNL and uninjured mice. After intraplantar injection of vehicle in SNL animals, the distance traveled and average velocity decreased and the time spent immobile increased compared to baseline testing before injection (Figure 7). These data indicate that these animals are hypersensitive after intraplantar injection, which is consistent with previous reports and data collected during mechanical threshold testing (Figure 4L). Diazoxide injection attenuated the reduction in (1) distance traveled (repeated measures ANOVA, F (2, 16) = 10.05, *p* < 0.05, Figure 7E) and (2) average velocity (repeated measures ANOVA, F (2, 16) = 11.23, *p* < 0.05, Figure 7G). Diazoxide also decreased the amount of time spent immobile compare to vehicle in SNL animals, although these results were not found to be significantly significant.

### 2.4. Genetic Ablation or Knockdown of SUR1 Increases Mechanical Hypersensitivity

Previously, mice with a global deletion of SUR1 have been shown to exhibit impaired glucose tolerance and significantly more hypoglycemia when fasted than control animals [17] but to date, there has been no data reporting any somatosensory changes in SUR1 KO animals. Here, SUR1 knock out mice tested for mechanical and thermal sensitivity showed significantly decreased mechanical thresholds compared to wild type mice (ANOVA, F(2, 64) = 3.85, *p* = 0.026; Tukey post-hoc test; 95% CI = 0.097 to 1.58; Figure 8A). Mice data were pooled, since no significant differences were detected across sexes (ANOVA, F(1, 66) = 1.24, *p* = 0.27) and across ages (Regression, *R* = 0.150, *R*^2^ = 0.022). Thermal thresholds were not altered in mice lacking SUR1 (Figure 8B).

Using a parallel strategy to impair SUR1 function specifically in the lumbar spinal cord and dorsal root ganglia, we introduced a shRNA via an adeno associated virus (AAV) to reduce *Abcc8* mRNA by intrathecal injection. Mechanical thresholds were reduced two weeks post injection and remained lower for at least four weeks after inoculation (repeated measures ANOVA, F(2, 64) = 3.85, *p* = 0.026; Tukey post-hoc test; 95% CI = 0.097 to 1.58; Figure 9A). Thermal thresholds were not altered 4 weeks after *Abcc8* shRNA administration (Figure 9B). Validation of viral inoculation was performed by fluorescence microscopy (Figure 9C) and reduction in *Abcc8* mRNA expression was confirmed via quantitative PCR (qPCR). The mRNA expression of *Abcc8* K_ATP_ channel subunits was decreased in dorsal root ganglia, spinal cord (paired t-test, *p* = 0.0046) and sciatic nerves (paired t-test, *p* = 0.045) five weeks after intrathecal injection (Figure 9D–F). The mRNA levels of other K_ATP_ channel subunits (i.e., *Abcc9*, *Kncj11* and *Kcnj8*) were not significantly altered after *Abcc8* shRNA inoculation (Appendix A).

### 2.5. Peripheral Application of SUR1 Subtype Agonists Decreased Evoked Mechanical Responses to Nerve Fibers Innervating the Hindpaw

We performed ex-vivo recordings from the sural nerve because: (1) recordings could be obtained from a receptive field innervated by the ligated nerve; and (2) the sural nerve innervates the lateral plantar surface of the hind paw and overlaps with the skin region that is tested by our mechanical and thermal paw withdrawal assays. Application of SUR1 subtype agonists provided the greatest analgesic effect during mechanical paw withdrawal, therefore the focus on the electrophysiology recordings was responsiveness to increasing mechanical force. This way we could directly compare our results from the mechanical paw withdrawal thresholds (Figure 4 and Figure 5) with the electrophysiological recordings. Mechanical responses were re-tested 15–20 min after removal of the reservoir ring following drug washout and these responses were largely returned to baseline (≥50% reversal of drug-induced response change, data not shown). Many units exhibited a large amount of variability in their evoked responses before diazoxide and glyburide application. To draw valid conclusions about whether a change in response level after drug application was due to the action of drug or to normal response variability, we normalized the data to the mechanical responsiveness before drug application (“post- pre”). Figure 10A gives an example of a C-fiber response before (Figure 10A) and after (Figure 10B) diazoxide application. Overall, diazoxide decreased C-fiber responses to increasing mechanical stimulation in SNL mice (F (5,70) = 3.70, *p* = 0.025); Tukey post-hoc comparing 0.1 g versus 5g and 8g; 95% CI = 3.05 to 28.2 and 0.55 to 25.7, respectively; Figure 10C, *n* = 8/group). Glyburide application did not significantly change the number of action potentials across mechanical forces. We also did not find any significant changes in number of action potentials post drug application in A delta-fibers (Figure 10D, *n* = 4) or A beta-fibers (Figure 10E, *n* = 6), however for several myelinated fibers there was a numerical decrease in responsiveness to supra threshold stimuli.

## 3. Discussion

The present study investigated the loss of SUR1-subtype K_ATP_ channels using a neuropathic pain model in mice. Intraplantar and intrathecal administration of SUR1-subtype agonists were beneficial for alleviating mechanical hypersensitivity and improve mobility after SNL. Administration of diazoxide, a SUR1 agonist, decreased evoked sural nerve responses to supratheshold mechanical stimulation. Genetic strategies confirmed that the loss of SUR1-subtype K_ATP_ channels contributes to mechanical hypersensitivity compared to control mice.

### 3.1. SUR1-Subtype K_ATP_ Channels Contribute to Analgesia in the Peripheral and Central Nervous System after Nerve Injury

Our qPCR data are largely in agreement with previous studies, indicating that K_ATP_ channel expression is decreased in DRG after nerve injury [12,18]. *Abcc8* (SUR1) and *Kcnj11* iso2 (Kir6.2) expression is significantly decreased in injured versus non-injured DRG (Figure 1) and similar changes were also seen in sciatic nerves. These results demonstrate that a loss of K_ATP_ channels appears to occur mostly in the peripheral nervous system, while loss in the spinal cord is less pronounced. The loss of *Kcnj11* (Kir6.2) was also detected in DRG and the spinal cord but appears to be isoform specific, as *Kcnj11* iso2 was much further decreased than *Kcnj11* iso1. To date, there are two confirmed *Kcnj11* isoforms in mice and humans. In our results, the second isoform is more robustly expressed in DRG than in the spinal cord and vice versa for the first isoform. It is unknown whether this baseline expression is similar between mice and humans or if changes seen after SNL in the mouse would be similar to humans.

Spinal nerve ligation in rats has been shown to decrease basal K_ATP_ activity in DRG and decrease opening probability after application of diazoxide, a SUR1-subtype K_ATP_ channel agonist [11]. In addition, diazoxide gradually enhanced, while glybenclamide inhibited, channel opening in SNL and control rat DRGs [11]. Our ex-vivo data are also in agreement with these previous findings—diazoxide decreased mechanical sensitivity to sural peripheral nerve fibers, whereas glyburide did not (Figure 10). Previous reports have indicated that K_ATP_ channel subunit expression is localized to different areas in the nervous system. For example, SUR1 is the predominant SUR subunit in DRG [12], Kir6.1 is expressed in the spinal cord but not in DRG and Kir6.2 is expressed in DRG but not the spinal cord [12,18]. Validation of these findings by immunohistochemistry and western blotting was attempted with current commercially available antibodies for SUR1 and SUR2 (NBP2-34077 and NBP1-84436 Novus Biologicals, sc-5789 and sc-25684 Santa Cruz Biotechnology); however, validation in SUR1 KO mice was unsuccessful (data not shown). Nonetheless, our mRNA expression data indicate that *Abcc8* (SUR1) is highly expressed in DRG and trigeminal ganglia and *Abcc9* (SUR2) is highly expressed in the spinal cord and brainstem (Appendix A).

### 3.2. SUR1 Involvement in Mechanical Hypersensitivity after Spinal Nerve Ligation

Our behavioral data also indicate that application of SUR1-subtype agonists are effective analgesics after local or intrathecal injection (Figure 4 and Figure 6). These changes were more apparent for mechanical threshold testing than for thermal paw withdrawal thresholds. These data are consistent with prior immunohistochemistry data indicating that SUR1 expression is decreased after SNL, especially in large myelinated fibers expressing NF200 [11,12]. Large neurons have been shown to be more likely to express K_ATP_ subunits, which might indicate a cell type-specific expression of K_ATP_ channels [9]. Mice lacking SUR1 displayed increased mechanosensitivity compared to their wild-type counterparts and genetic knockdown of SUR1 using AAV viral vectors also decreased mechanosensitivity compared to their control virus counterparts. These data collectively demonstrate that a loss of K_ATP_ channel activity is attributed to mechanical hypersensitivity. These data are particularly intriguing from a pharmaceutical perspective, as many individuals with neuropathic pain report mechanical hypersensitivity or mechanical allodynia [19]. The open field testing data indicated that application of K_ATP_ channel agonists to the periphery is able to increase behavioral activity after SNL but not in un-injured animals. These data are important, as increased locomotor activity are noted in rodent studies and with a neuronal gain-of-function mutation of Kir6.2 [20]. The ex-vivo recordings suggest that SUR1-targeting agonists were able to significantly decrease mechanically-evoked responses to C-fibers but not A delta-or A beta-fibers. A more detailed follow-up study could demonstrate if responsiveness to K_ATP_ channel agonists is dependent on subclasses of myelinated fibers [21]. Interestingly, the Kir6.2/SUR2 agonist, levcromakalim, also elicited a slight analgesic effect when delivered via hindpaw (Figure 4G) or intrathecally (Figure 6E). This data indicates that agonists that target vascular K_ATP_ channels may enhance vasodilation and improve neuropathy symptoms as seen with other vasodilatory agents [22,23,24].

SUR1 expression is not restricted to the central or peripheral nervous system. Global deletion of SUR1 results in impaired glucose tolerance due to loss of beta cell activity in the pancreas [17]. Fed glucose levels were not found to be significantly different across genotypes in this study (data not shown), which is in agreement from previous studies but in contrast with others [25]. We conclude that any changes in blood glucose as a result of SUR1 global deletion are minimal and probably do not contribute to the mechanical hypersensitivity found in behavioral assays. We can assume this, because the targeted deletion of *Abcc8* using shRNA intrathecal injection also resulted in similar mechanical threshold findings, indicating that loss of SUR1 in the spinal cord and dorsal root ganglion are more likely responsible for this behavioral phenotype than changes in glucose metabolism. In the spinal cord, SUR1 has also been found to form stable co-association with TRPM4 channels, a monovalent cation channel that is sensitive to calcium [26]. These SUR1-regulated NCCa-ATP channels have been found to be involved in cerebral ischemia, traumatic brain injury, spinal cord injury and subarachnoid hemorrhage [27,28,29]. Although activation of SUR1 expressing K_ATP_ channels may be beneficial for pain due to neuropathy or inflammation [30,31,32], pain due to hypoxia or edema of the central nervous system may respond negatively to these treatments. In addition to neurons, SUR1 subunits are found in central nervous system astrocytes [33], DRG satellite cells [12] and brain microglia [34]. Our current data do not segregate the contribution(s) of different cell types in the involvement of hyperalgesia after SNL; however, future studies could carefully dissect the role of neurons versus glia through cre-lox recombination or viral vector strategies utilizing different cell-type promotors in the nervous system.

In conclusion, our study provides further substantiated evidence that loss of K_ATP_ channel expression can contribute to mechanical hypersensitivity seen during peripheral nerve injury. Specifically, changes in SUR1 and *Kcnj11* K_ATP_ channel subunit expression after SNL were largely found in the peripheral nervous system. These data suggest that the activity of SUR1—and possibly Kir6.2—are important for the development of mechanical hyperalgesia seen during peripheral neuropathies.

## 4. Material and Methods

### 4.1. Animals and Breeding

All experimental procedures involving animals were performed and approved in accordance with the University of Minnesota Institutional Animal Care and Use Committee guidelines. Adult C57Bl6 mice were obtained via Charles River (5–6 weeks old, Raleigh, NC). SUR1 global knock out mice (SUR1 KO) were obtained from the laboratory of Dr. Joseph Bryan at the Pacific Northwest Research Institute (Seattle, WA, USA) as used in previous studies [17] and kept on a C57Bl6 background. Verification of genotype was done via DNA extraction from mouse tail biopsies by the “Hot Shot” method [35] and DNA amplification using Amplitaq Gold DNA Polymerase (ThermoFisher Scientific, Waltham, MA, USA). The presence of the SUR1 KO allele was confirmed using PCR with the following primer set: forward 5′- CTG TCC ATC TGC ACG AGA CT-3′ and reverse 5′-AGG TTG TTG GTG GAG GTC AG -3′ yielding a KO band of 350 bp and a WT allele band of 524 bp [36]. The temperature and cycling protocols were repeated for 40 cycles as follows: 95 °C for 10 min, 94 °C for 30 s, 62 °C for 30 s and 72 °C for 45 s.

### 4.2. Spinal Nerve Ligation

To create the neuropathic pain model, spinal nerve ligation (SNL) was performed on adult C57Bl6 mice [37]. The SNL surgery was performed, as previously described, on mice under deep isoflurane anesthesia and was completed within 20 min [38]. Using aseptic technique, an incision over L5-S1 was made, the left L6 transverse process was partially removed and the left L5 nerve was isolated and transected (ipsilateral). The right spinal nerve was left intact as a control (contralateral).

### 4.3. Tissue Collection and RNA Extraction

Mice were anesthetized with 5% isoflurane in oxygen and decapitated. In C57Bl6 mice, SNL and un-injured mice, cerebral cortex (brain), brainstem, trigeminal ganglia, sciatic nerves and lumbar dorsal root ganglia and spinal cords (L3–L6 segments) were collected. All tissues were dissected and snap frozen in liquid nitrogen. Tissues were stored at −80 °C for long-term storage. RNA was purified from the frozen tissue samples using Trizol regent (Sigma-Aldrich, St. Loius, MO, USA). RNA was isolated using the RNeasy Mini Spin Columns (Qiagen, Germantown, MD, USA) with a DNase I treatment step. The quantity and quality of the eluted RNAs were assessed by Nanodrop spectrophotometry (Thermo Fisher Scientific) and/or Qubit fluorometric quantification (Thermo Fisher Scientific).

### 4.4. Quantitative PCR

Synthesis of cDNA from all purified RNA samples (50 ng) was performed using the Omniscript RT Kit and protocol from Qiagen. Primers were designed using sequences from NCBI primer design (Table 1). qPCR was performed with LightCycler 480 technology (Roche, Branchburg, NJ, USA) and SYBR Green I dye (Roche). Gene specific cDNA copy number was typically quantified against a ≥5 point, 10 fold serial dilution of gene specific cDNA standard in the range of 1e6 to 1e1 copies per reaction. The temperature and cycling protocol were as follows: 95 °C for 10 sec, 60 °C for 10 sec and 72 °C for 20 sec and was repeated for 45 cycles. Expression of a housekeeping gene, *18S*, was used as an internal control. The presence of genomic DNA was assessed using negative RT-PCR controls. Relative expression levels of each gene of interest were determined by: (mean concentration)/(mean concentration of *18S*), for each sample calculated using LightCycler^®^ 480 Software. Two to three experimental replicates were performed for each sample.

### 4.5. Evoked Behavioral Experiments

Mice were acclimatized to the testing platforms/arenas by placement in a transparent plastic box and were allowed to acclimate to the test chamber for at least 20 min. All behavioral tests were conducted in a blinded manner [39]. For mechanical studies, withdrawal latencies were assessed using a hand held electronic Von Frey hair (2392, IITC Life Sciences, Woodland Hills, CA, USA) on a wire mesh floor. Thermal latencies were assessed using a radiant paw withdrawal assay (390G, IITC Life Sciences, Woodland Hills, CA, USA) on a feedback controlled glass plate set to 30 °C. Positive responses included an abrupt flinching/withdrawal of the hind paw from the stimulus [40]. For baseline sensitivity, threshold measurements were run in at least triplicate on the left and right paw, with at least a 30 sec inter stimulus interval between paws. After drug injection, thresholds and latencies were measured on both paws 3, 15, 30, 45 and 60 min after administration. Behavioral tests in SNL animals were performed through postoperative day 14–60. Behavioral hypersensitivity to mechanical and thermal stimuli lasting 8 weeks or greater after nerve injury in rodents has been shown in previous studies [41,42,43]. Any animals that appeared to have altered baseline activity/responses in comparison to initial testing were removed from the study (less than 5% of animals).

### 4.6. Open Field Testing

Animals were placed in the center of a 40 cm × 40 cm open field arena and activity was recorded with a camcorder for offline analysis (HDR-CX405; Sony Corp., Tokyo, Japan). A digital light meter was used to verify equal light distribution across the arena (50–60 lux). For baseline recordings, animals were allowed to explore for 15 min. After drug injection, animals were returned to the arena and allowed to explore for 30 additional minutes. The behavioral parameters were scored by the software Ethowatcher^®^ (developed by the Laboratory of Comparative Neurophysiology of the Federal University of Santa Catarina, freely available on www.ethowatcher.ufsc.br, IEB-UFSC [44]), which translated recorded movement into numerical data for frame-by-frame analysis. Behavioral parameters for distance traveled, time spent immobile, average velocity and change in angular direction were compared across treatment groups.

### 4.7. Drug Delivery

Diazoxide, Pinacidil, NN-414, Nicorandil, Levcromakalim (gift from Peter Dosa, Department of Medicinal Chemistry, University of Minnesota), Tolbutamide, Gliclazide and Glyburide (Sigma Aldrich or Tocris, Minneapolis, MN, USA) were initially diluted in DMSO and further diluted with saline to a final concentration of 10–100 uM in 5% DMSO in saline (vehicle). Compounds used in this study were used at concentrations found in previous behavioral studies [45,46,47,48,49]. Intraplantar injections were performed by gently restraining the animal in a Plexiglas restrainer or decapicone. A 27–33 gauge needle was inserted into the hindpaw skin at a 15–30° angle and 10 uL of solution was slowly delivered. A successful injection was determined when a small bleb was seen [50]. Intraperitoneal injections were delivered into the lower left abdominal quadrant using a 27 gauge needle at a volume of 100 μL. Intrathecal injections (10 μL) were performed by direct lumbar puncture in awake mice as previously described [51,52]. Most mice were tested with different chemicals, with at least 7 days between successive tests of active chemicals to avoid carryover effects.

### 4.8. shRNA Delivery

In order to achieve efficient and long-lasting knockdown of SUR1 K_ATP_ channel subunits in the spinal cord and dorsal root ganglia, we utilized intrathecal injection of short hairpin (shRNA)-encoding AAV9 viral vectors [53]. The viral constructs AAV9-GFP-U6-m-*ABCC8*-shRNA and AAV9-GFP-U6-scramble-shRNA (shAAV-251792 and 7045, titer: 10^13 gc/mL, in PBS with 5% glycerol, Vector Biolabs, Malvern, PA, USA) were stored at −80 °C until injection. AAV9-viruses were injected intrathecally in awake mice as described previously [54,55,56] and behavioral assessments were performed one to four weeks post injection. Verification of mRNA knockdown was achieved using qPCR of harvested lumber spinal cords and dorsal root ganglia and sciatic nerves (see ‘Quantitative PCR’). Histological sections were taken from some animals in order to demonstrate that the delivery of AAV vectors within the lumbar intrathecal space leads to expression of green fluorescent protein (GFP) in spinal cord, DRGs and peripheral nerves. Sections (10 uM, Leica CM3050) were mounted onto electrostatically charged slides and images were collected using a Nikon TiS Microscope and associated software.

### 4.9. Ex-Vivo Electrophysiology

Mice were euthanized with 5% isoflurane in oxygen and cervical dislocation or decapitation. The hairy skin of the hind paw, together with the sural nerve, was carefully dissected and transferred to an in vitro system described previously [38,57]. The skin was mounted corium side up in an organ bath and superfused (10 mL/min) with synthetic interstitial fluid (SIF) heated to ~32 °C and continuously bubbled with 95% oxygen/5% carbon dioxide to obtain a SIF pH of 7.4. Under a microscope, the nerve was teased into smaller bundles that were placed onto a recording electrode to achieve single-fiber activity. After localizing the receptive field of the fiber with a glass probe, electrical stimuli was applied at the receptive field with a concentric electrode to calculate conduction velocity (CV). Fibers with a CV of less than 1 m/s were classified as C-fibers, those with a CV between 1 and 10 m/s were classified as Aδ-fibers and those with a CV more than 10 m/s were classified as Aβ-fibers [58]. Mechanical sensitivity was assessed by applying an ascending series of suprathreshold mechanical stimuli (0.1 to 8 g or 0.98 to 78.5 mN for 2 s, separated by 1 min) to the receptive field with a blunt probe (1-mm diameter) connected to a computer-controlled mechanical stimulator (model 305C; Aurora Scientific, Aurora, ON, Canada) [59]. After baseline mechanical and thermal testing was completed, SIF within the ring was replaced by a 100 µM diazoxide or glyburide SIF solution, which was maintained for 5 min, followed by repeated mechanical stimulation. Action potentials were filtered, amplified, digitized and stored on a computer using DAPSYS, which was also used to control the mechanical stimulator and record bath temperature (Brian Turnquist, Bethel University, St. Paul, Minnesota; http://www.dapsys.net). The total number of evoked action potentials to the mechanical stimulus, including a post-stimulus period of 3 s, was used for data analysis pre- and post-drug application.

### 4.10. Statistical Analysis

We performed the appropriate t-test or one-way, two-way or repeated measures ANOVA followed by *post hoc* analysis (as indicated) to determine the significance of treatment groups for gene expression, mechanical and thermal paw withdrawals, open field testing parameters and ex-vivo single-unit recordings. All statistical analyses were carried out with Prism 6.0 (GraphPad Software Inc., San Diego, CA, USA) or SPSS (Statistics 23, IBM, Chicago, IL, USA). The data are presented as means ± standard error and *p* < 0.05 was considered statistically significant.

## Figures and Tables

**Figure 1 ijms-20-02251-f001:**
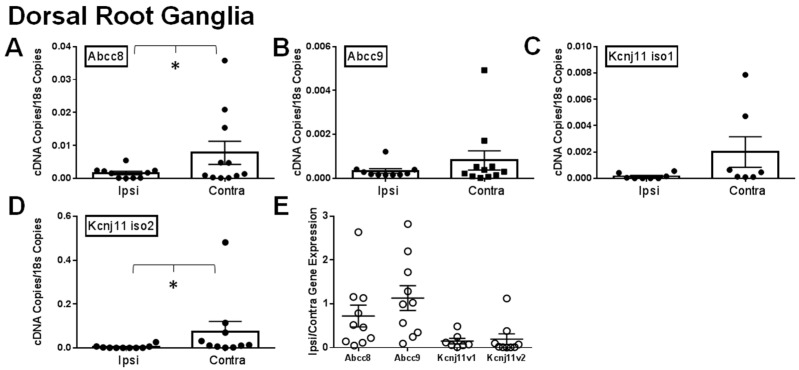
Expression of K_ATP_ channel subunits in dorsal root ganglia are decreased after spinal nerve ligation (SNL). Relative expression of K_ATP_ channel subunits in ipsilateral (Ipsi) and contralateral (Contra) dorsal root ganglia >30 days post SNL. Relative gene expression for *Abcc8* (protein: SUR1, **A**), *Abcc9* (protein: SUR2, **B**), *Kcnj11* iso1 (protein: Kir6.2, **C**) and *Kcnj11* iso2 (protein: Kir6.2, **D**) compared against matching tissue expression for *18s*. * *p* < 0.05 comparison between *Ipsi* vs. *Contra*. (**E**). Data in A–D are expressed as mean fold change (Ipsi/Contra) between matching samples.

**Figure 2 ijms-20-02251-f002:**
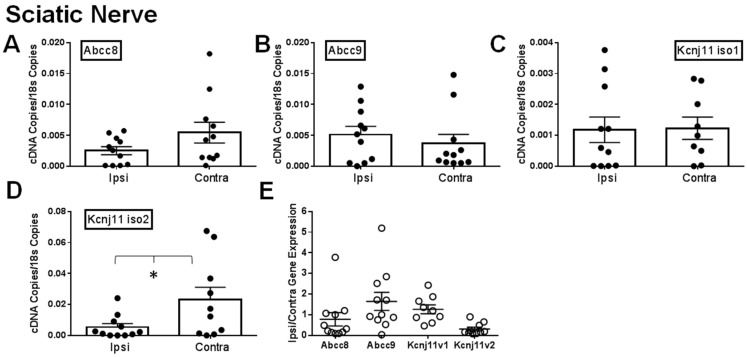
Expression of K_ATP_ channel subunits in sciatic nerves are decreased after spinal nerve ligation (SNL). Relative expression of K_ATP_ channel subunits in ipsilateral (Ipsi) and contralateral (Contra) sciatic nerves >30 days post SNL. Relative gene expression for *Abcc8* (protein: *SUR1*, **A**), *Abcc9* (protein: SUR2, **B**), *Kcnj11* iso1 (protein: Kir6.2, **C**) and *Kcnj11* iso2 (protein: Kir6.2, **D**) compared against matching tissue expression for *18s*. * *p* < 0.05 comparison between *Ipsi* vs. *Contra*. (**E**). Data in A–D are expressed as mean fold change (Ipsi/Contra) between matching samples.

**Figure 3 ijms-20-02251-f003:**
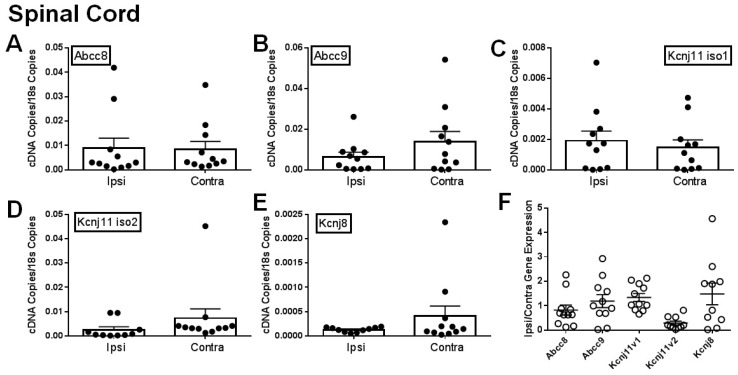
Expression of K_ATP_ channel subunits in spinal cord are decreased after spinal nerve ligation (SNL). Relative expression of K_ATP_ channel subunits in ipsilateral (Ipsi) and contralateral (Contra) spinal cords 30 days post SNL. Relative gene expression for *Abcc8* (protein: SUR1, **A**), *Abcc9* (protein: SUR2, **B**), *Kcnj11* iso1 (protein: Kir6.2, **C**), *Kcnj11* iso2 (protein: Kir6.2, **D**) and *Kcnj8* (protein: Kir6.1, **E**) compared against matching tissue expression for *18s*. * *p* < 0.05 comparison between *Ipsi* vs. *Contra*. (**F**). Data in A-D are expressed as mean fold change (Ipsi/Contra) between matching samples.

**Figure 4 ijms-20-02251-f004:**
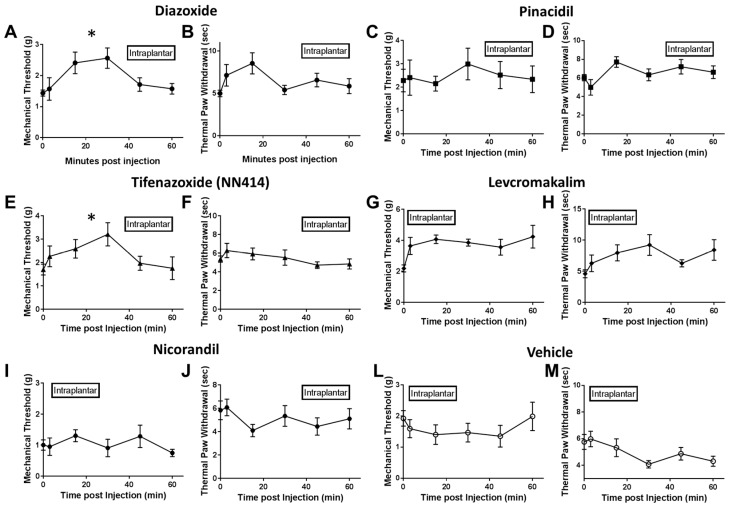
Paw withdrawal thresholds and latencies are increased after intraplantar administration of SUR1 agonists. Administration of the SUR1 agonist diazoxide (**A**) and the Kir6.2/SUR1 agonist NN-414 (**E**) significantly increased mechanical paw withdrawal thresholds compared to vehicle (**L**) (10 μL, hindpaw; repeated measures ANOVA, F (5, 44) = 9.719, * *p* < 0.05, repeated measures ANOVA, CI diazoxide = −2.999 to −0.9227, CI NN414 = −1.167 to −0.6546). Thermal latencies for diazoxide (**B**) and NN-414 (**F**) were not significantly different than vehicle (**M**). Mechanical and thermal thresholds for the SUR2 agonist pinacidil (**C**,**D**), the Kir6.2/SUR2 agonist levcromakalim (**G**,**H**) or the Kir6.2/SUR2B agonist nicorandil (**I**,**J**) were not significantly changed compared to vehicle (**L**,**M**). Error bars: SEM; *n* = 5–12/group.

**Figure 5 ijms-20-02251-f005:**
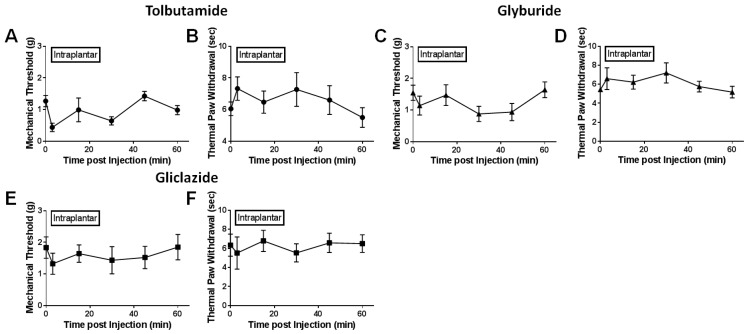
Paw withdrawal thresholds and latencies do not change after intraplantar administration of K_ATP_ channel antagonists. Administration of the SUR2 antagonist Tolbutamide (**A**,**B**), the SUR1 and SUR2 antagonist glyburide (**C**,**D**) or the SUR1 antagonist Gliclazide (**E**,**F**). Mechanical and thermal thresholds were not significantly changed compared to vehicle (Figure 4). Error bars: SEM; *n* = 4–10/group.

**Figure 6 ijms-20-02251-f006:**
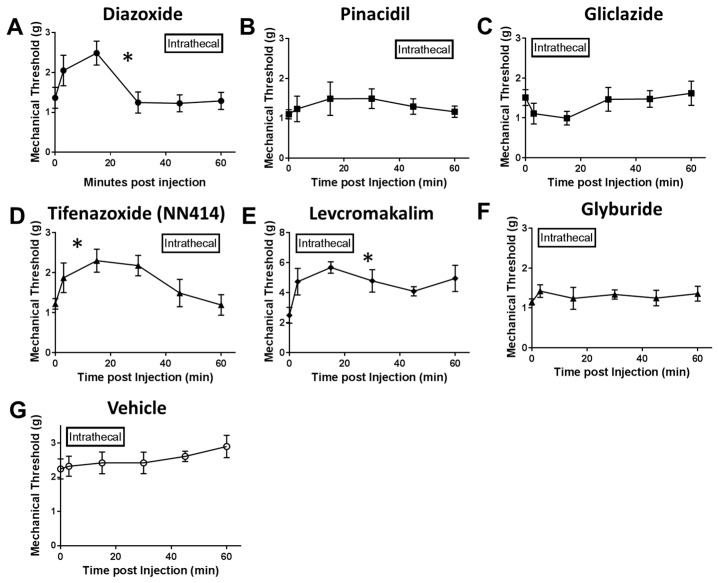
Paw withdrawal thresholds and latencies are increased after intrathecal administration of SUR1 agonists. Administration of the SUR1 agonist diazoxide (**A**), the Kir6.2/SUR1 agonist NN-414 (**D**) or the Kir6.2/SUR2 agonist levcromakalim (**E**) significantly increased paw withdrawal thresholds compared to vehicle (**G**) (10 μL, hindpaw; * *p* < 0.05, repeated measures ANOVA, F (6, 56) = 32.94, * *p* < 0.05, CI diazoxide = −0.7470 to −0.4126, CI NN414 = −0.999 to −0.1607, CI levcromakalim = −3.363 to −1.969). Error bars: SEM; *n* = 5–10/group.

**Figure 7 ijms-20-02251-f007:**
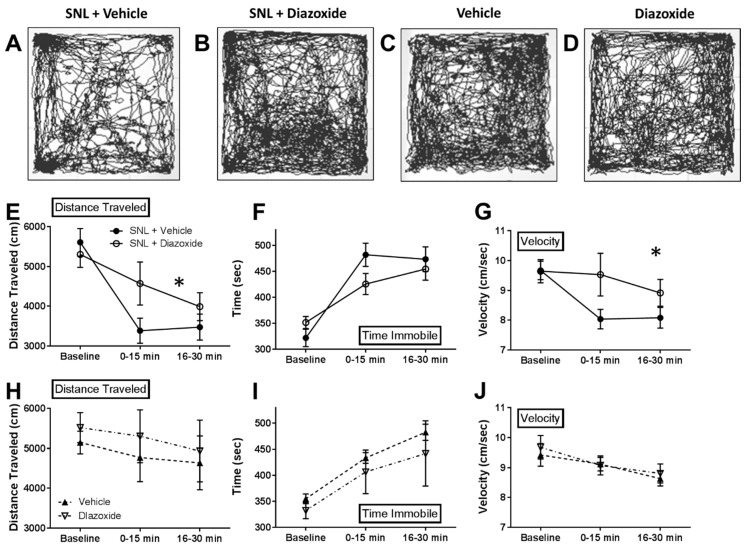
Intraplantar diazoxide increases mobility in mice after SNL. Open field testing (40 × 40 cm arena) used to track mouse movements before (15 min) and after (30 min total) intraplantar injection with saline (**A**) or diazoxide (**B**, 100 μM) after SNL or in uninjured mice (**C**,**D**). (**E**) Significant increase in distance traveled for diazoxide treated mice versus saline treated SNL mice (repeated measures ANOVA, F (2, 16) = 10.05, *p* < 0.05). (**F**) Time spent immobile was not significantly different between vehicle treated and diazoxide treated SNL animals. (**G**) Significant increase in average velocity for diazoxide treated mice versus saline treated SNL mice (repeated measures ANOVA, F (2, 16) = 11.23, *p* < 0.05). No significant changes in mobility were found in uninjured mice after vehicle or diazoxide injection (**H**–**J**). *n* = 5 for all groups.

**Figure 8 ijms-20-02251-f008:**
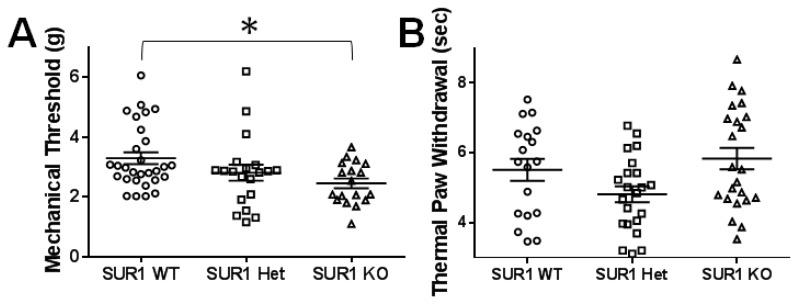
Mechanical paw withdrawal thresholds but not thermal paw withdrawal thresholds are decreased in male and female SUR1 KO mice compared to WT mice. (**A**) Mechanical paw withdrawal thresholds are significantly decreased in SUR1KO mice compared to WT mice (ANOVA, F(2, 64) = 3.85, *p* = 0.026; Tukey post-hoc test; 95% CI = 0.097 to 1.58). (**B**) Thermal withdrawal latencies were not significantly different between genotypes. Data across sex and age of age are aggregated since no significant differences were found.

**Figure 9 ijms-20-02251-f009:**
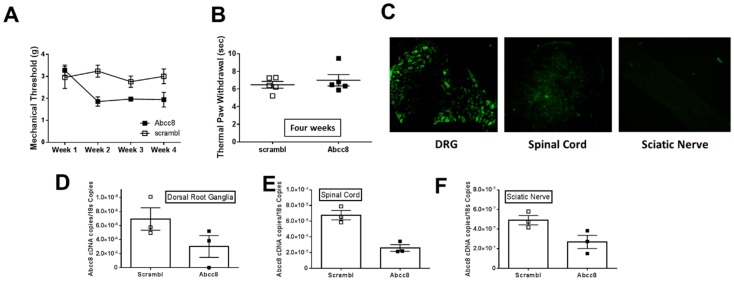
Intrathecal injection of *Abcc8* shRNA decreases mechanical paw withdrawal latencies in mice. (**A**) Intrathecal injection of *AAV9-GFP-U6-m-ABCC8-shRNA* (10 μL) significantly decreases mechanical paw withdrawal thresholds over time in mice compared to control injections (AAV9-GFP-U6-m-scramble-shRNA) (repeated measures ANOVA, F (3, 24) = 3.02, *p* = 0.049). (**B**) Thermal paw withdrawals were not significantly changed four weeks post injection. (**C**) Histological images of dorsal root ganglia (DRG), spinal cord and sciatic nerve section from an animal after Abcc8 shRNA injection. Green fluorescent protein is detected in both primary afferent cell bodies and in the spinal cord after five weeks post injection. Images taken at 10× magnification. The mRNA expression of *Abcc8* K_ATP_ channel subunits are decreased in (**D**) dorsal root ganglia, (**E**) spinal cord (paired t-test, *p* = 0.0046) and (**F**) sciatic nerves (paired t-test, *p* = 0.045), five weeks after intrathecal injection.

**Figure 10 ijms-20-02251-f010:**
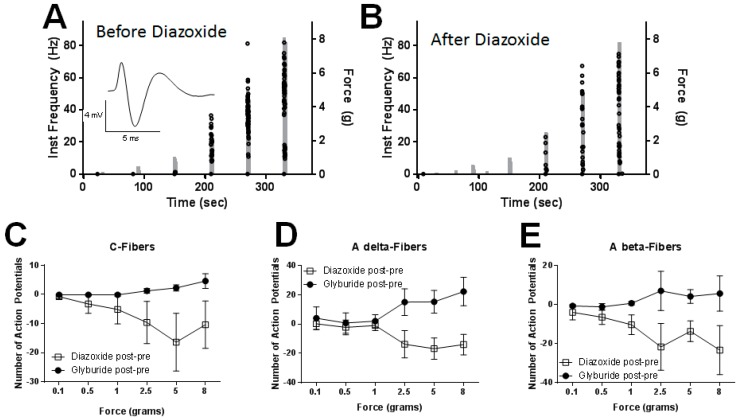
The SUR1 agonist, diazoxide, decreases mechanical responsiveness of unmyelinated, C-fibers. (**A**) Increase in number (dots) and peak instantaneous frequency (left *y*-axis) of C-fiber action potentials as the force of stimulation is increased in a step-wise fashion (right *y*-axis). Inset: Example trace from recording. (**B**) Application of diazoxide (100 uM, 10 min) to the receptive field of a C-fiber decreases the number of action potentials compared to before application. (**C**) Diazoxide decreases C-fiber responses to increasing mechanical stimulation in SNL mice (F (5,70) = 3.70, *p* = 0.025); Tukey post-hoc comparing 0.1 g versus 5 g and 8 g; 95% CI = 3.05 to 28.2 and 0.55 to 25.7, respectively; *n* = 8/group). Glyburide application did not significantly change the number of action potentials across mechanical forces. No significant changes in number of action potentials post drug application were found in myelinated A delta-fibers (**D**) or A beta-fibers (**E**). Data presented as number of action potentials after drug application subtracted from the number of action potentials during the same time period before drug application.

**Table 1 ijms-20-02251-t001:** Primers used in quantitative PCR studies.

Gene	Accession Number	Protein	Forward	Reverse
*Abcc8*	NM_011510.3	SUR1	CCAACACGAGCCTTGAACTT	AGGTTGTTGGTGGAGGTCAG
*Abcc9*	NM_011511.2	SUR2	TGTAGGCCAAGTGGGTTGTG	TCTGCTTCGGGTTGCTTCAA
*Kcnj11* iso1	NM_010602.3	Kir6.2 isoform 1	CGCCCACAAGAACATTCGAG	GCAGAGTGTGTGGCCATTTG
*Kcnj11* iso2	NM_001204411.1	Kir6.2 isoform 2	ACCACGTCATCGACTCCAAC	TGGTTTCTACCACGCCTTCC
*Kcnj8*	NM_008428.5	Kir6.1	GGCACCATGGAGAAGAGTGG	CAAAACCGTGATGGCCAGAG
Rn*18s*	NR_003278.3	*18s*	CGCCGCTAGAGGTGAAATTCTT	CAGTCGGCATCGTTTATGGTC

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
