# Peer review of "Modulation of SUR1 KATP Channel Subunit Activity in the Peripheral Nervous System Reduces Mechanical Hyperalgesia after Nerve Injury in Mice"

_ijms, 2019, doi:10.3390/ijms20092251_

Round 1

Reviewer 1 Report

This is a very comprehensive study concerning the role of SURI KATP channel in the modulation of neuropathic pain in sciatic nerve ligation model. The author either used the SUR1 KATP agonist or utility of genetic knock down by associated adenovirus vector to assess the outcome of neuropathic pain. However, nearly all of data were presented by the RT-PCR. The cell identity in SUR1 KATP after nerve ligation in the nerve, dorsal root ganglion and dorsal spinal cord were not determined.

The authors should add one immunofluorescent imaging of SUR1 KATP merged with NF-200/S-100 in the sciatic nerve, Neu-N in the dorsal root ganglion and dorsal spinal cord in the nerve ligation animals subjected to SUR1 KATP agonist and intrathecal injection of short hairpin (shRNA)-414 encoding AAV9 viral vectors.

Author Response

This is a very comprehensive study concerning the role of SURI KATP channel in the modulation of neuropathic pain in sciatic nerve ligation model. The author either used the SUR1 KATP agonist or utility of genetic knock down by associated adenovirus vector to assess the outcome of neuropathic pain. However, nearly all of data were presented by the RT-PCR. The cell identity in SUR1 KATP after nerve ligation in the nerve, dorsal root ganglion and dorsal spinal cord were not determined.

Point 1:  The authors should add one immunofluorescent imaging of SUR1 KATP merged with NF-200/S-100 in the sciatic nerve, Neu-N in the dorsal root ganglion and dorsal spinal cord in the nerve ligation animals subjected to SUR1 KATP agonist and intrathecal injection of short hairpin (shRNA)-414 encoding AAV9 viral vectors.

The authors chose not to provide immunofluorescent slides of SUR1 co-localized with markers for NF200 or Neu-N for two reasons:  (1) These studies have been done previously (Kawano et al., 2009; Zoga et al., 2010; Wu et al, 2011) and indicate that expression of SUR1 is present in large myelinated and small unmyelinated neurons, and decreases after nerve injury models in rodents and (2)  The use of antibodies to co-localize for SUR1 for immunohistochemistry is questionable at best.  We have elaborated our troubles using and validating commercially available primary antibodies in the discussion section (lines 278-282).  Since we are not confident in the ability of our antibodies to identify SUR1 or SUR2 positive cells, we chose to not include this data.

We have added some images from the GFP-tagged AAV9 viral vectors to indicate that expression of the vector, under a global promotor, is present in many fibers and cells of the peripheral and central nervous system five weeks post injection (Figure 9C). 

Reviewer 2 Report

Dear Editor,

The work submitted by Dr Luu entitled ‘Modulation of SUR1 KATP channel subunit activity in the peripheral nervous system reduces mechanical hyperalgesia after nerve injury in mice’ demonstrated that expression levels of a specific subtype of ATP-sensitive K+ channel (SUR1) in the dorsal root ganglia (but not spinal cord or sciatic nerve) inversely correlated with the occurrence of mechanical hyperalgesia in a model of peripheral nerve injury.

Through pharmacological, global knockout mice and local (intrathecal) AAV-mediated gene silencing strategies, the authors initially show the following:

1. That the SUR1 agonist (Diazoxide) decreased mechanically evoked hypersensitivity, hence reduced the effects of experimentally-induced neuropathy. Similarly, the dual SUR1-Kir6.2 agonist (NN414) also ameliorated mechanical hyperalgesia, which interestingly, was not impacted in animals treated with SUR1 antagonists.

The authors report no effects of SUR1 agonist treatment on thermal hyperalgesia.

2. SUR1 knockout mice exhibit lower mechanical threshold as compared with WTs.

3. Intrathecal AAV-gene silencing also increased mechanical hyperalgesia but not thermal hyperalgesia and reduced the spinal, DRG and sciatic nerve expression of abcc8 gene (SUR1).

The authors also test different routes of administration (intraplantar or intrathecal), with the idea of testing whether the treatment acted peripherally or centrally and found that treatment through both routes effectively restored sensorial disfunctions.   

Finally, open field testing is utilised to assess the distance travelled by nerve injured animals before and after intraplantar injection of the SUR1 agonist diazoxide, showing that diazoxide improves both distance and speed, here used as indirect indicators of reduced neuropathy-induced hyperalgesia.

In general, the experiments have been well-structured and provide robust evidence to support the use/development of drugs targeting the SUR1 KATP channel subunit to ameliorate an important debilitating component of neuropathy (mechanical hyperalgesia). Nonetheless, this reviewer has some concerns over the use of appropriate controls in the open field testing.

Indeed, it would have been more convincing if the authors included the tracked area and relative measures for uninjured control injected with saline (vehicle-control) and uninjured controls injected with diazoxide (drug-treatment control) in addition to the ones reported. This would have ruled out any potential confounder, including possible central effects of the drug on locomotor behaviour and/or would have shown the amplitude of the effect of spinal nerve ligation on locomotion. This concern is based on a previous work showing that gain-of-function mutations in Kir6.2 leads to enhanced locomotor activity and exploratory behaviour using the Open field testing (Lahmann et al., 2014; Physiol Behav. 2014 Apr 22;129:79-84).

Author Response

The work submitted by Dr Luu entitled ‘Modulation of SUR1 KATP channel subunit activity in the peripheral nervous system reduces mechanical hyperalgesia after nerve injury in mice’ demonstrated that expression levels of a specific subtype of ATP-sensitive K+ channel (SUR1) in the dorsal root ganglia (but not spinal cord or sciatic nerve) inversely correlated with the occurrence of mechanical hyperalgesia in a model of peripheral nerve injury.

Through pharmacological, global knockout mice and local (intrathecal) AAV-mediated gene silencing strategies, the authors initially show the following:

1. That the SUR1 agonist (Diazoxide) decreased mechanically evoked hypersensitivity, hence reduced the effects of experimentally-induced neuropathy. Similarly, the dual SUR1-Kir6.2 agonist (NN414) also ameliorated mechanical hyperalgesia, which interestingly, was not impacted in animals treated with SUR1 antagonists.

The authors report no effects of SUR1 agonist treatment on thermal hyperalgesia.

2. SUR1 knockout mice exhibit lower mechanical threshold as compared with WTs.

3. Intrathecal AAV-gene silencing also increased mechanical hyperalgesia but not thermal hyperalgesia and reduced the spinal, DRG and sciatic nerve expression of abcc8 gene (SUR1).

The authors also test different routes of administration (intraplantar or intrathecal), with the idea of testing whether the treatment acted peripherally or centrally and found that treatment through both routes effectively restored sensorial disfunctions.   

Finally, open field testing is utilised to assess the distance travelled by nerve injured animals before and after intraplantar injection of the SUR1 agonist diazoxide, showing that diazoxide improves both distance and speed, here used as indirect indicators of reduced neuropathy-induced hyperalgesia.

In general, the experiments have been well-structured and provide robust evidence to support the use/development of drugs targeting the SUR1 KATP channel subunit to ameliorate an important debilitating component of neuropathy (mechanical hyperalgesia). Nonetheless, this reviewer has some concerns over the use of appropriate controls in the open field testing.

Indeed, it would have been more convincing if the authors included the tracked area and relative measures for uninjured control injected with saline (vehicle-control) and uninjured controls injected with diazoxide (drug-treatment control) in addition to the ones reported. This would have ruled out any potential confounder, including possible central effects of the drug on locomotor behaviour and/or would have shown the amplitude of the effect of spinal nerve ligation on locomotion. This concern is based on a previous work showing that gain-of-function mutations in Kir6.2 leads to enhanced locomotor activity and exploratory behaviour using the Open field testing (Lahmann et al., 2014; Physiol Behav. 2014 Apr 22;129:79-84).

The authors appreciate this comment and this citation, which we have now added to the discussion (lines 312-315).  The requested control data has been added to Figure 7.  Since these injections were done intradermally into the plantar hindpaw, a central effect was not observed, as predicted. 

Round 2

Reviewer 1 Report

The authors tried to response to comments and made the changes as possible. The paper should be published.  

Reviewer 2 Report

Dear Editor,

The manuscript has been substantially reviewed and most of the issue raised by this and other reviewers have been addressed. I have no further comments on the quality of the manuscript, which has significantly improved.